# Dual Roles of Plasma miRNAs in Myocardial Injuries After Polytrauma: miR-122-5p and miR-885-5p Reflect Inflammatory Response, While miR-499a-5p and miR-194-5p Contribute to Cardiomyocyte Damage

**DOI:** 10.3390/cells14040300

**Published:** 2025-02-18

**Authors:** Jiaoyan Han, Liudmila Leppik, Larissa Sztulman, Roberta De Rosa, Victoria Pfeiffer, Lewin-Caspar Busse, Elena Kontaxi, Elisabeth Adam, Dirk Henrich, Ingo Marzi, Birte Weber

**Affiliations:** 1Department of Trauma Surgery and Orthopedics, University Hospital, Goethe University Frankfurt, 60590 Frankfurt, Germany; leppik@med.uni-frankfurt.de (L.L.); sztulman@med.uni-frankfurt.de (L.S.); caspar.busse@gmail.com (L.-C.B.); elena.kontaxi@unimedizin-ffm.de (E.K.); d.henrich@trauma.uni-frankfurt.de (D.H.); marzi@trauma.uni-frankfurt.de (I.M.); bi.weber@med.uni-frankfurt.de (B.W.); 2Department of Cardiology, University Hospital, Goethe University Frankfurt, 60590 Frankfurt, Germany; robertaderosa984@gmail.com (R.D.R.); v.pfeiffer@med.uni-frankfurt.de (V.P.); 3Department of Anaesthesiology, Intensive Care Medicine and Pain Therapy, University Hospital, Goethe University Frankfurt, 60590 Frankfurt, Germany; adam@med.uni-frankfurt.de

**Keywords:** polytrauma, cardiac damage, miRNA, troponin T, NLRP3, caspase, next-generation sequencing

## Abstract

Cardiac injury after severe trauma is associated with higher mortality in polytrauma patients. Recent evidence suggests that miRNAs play a key role in cardiac pathophysiology and could serve as potential markers of cardiac damage after polytrauma. To explore this hypothesis, plasma miRNA profiles from polytrauma patients (ISS ≥ 16) with and without cardiac injury, stratified by troponin T levels (TnT, > 50 pg/mL vs. < 12 pg/mL), were analysed using NGS and validated via RT-qPCR. Five miRNAs (miR-122-5p, miR-424-5p, miR-885-5p, miR-194-5p, and miR-499a-5p) were found to be significantly upregulated in polytrauma patients with elevated TnT levels. miR-122-5p was associated with markers of right ventricular dysfunction (TAPSE) and left ventricular hypertrophy (IVS/LVPW), while miR-885-5p correlated with left ventricular hypertrophy (IVS/LVPW) and diastolic dysfunction (E/E’ ratio). In vitro, miR-194-5p mimic and miR-499a-5p mimic exhibited more active roles in cardiomyocyte injury by increasing caspase-3/7 activity and/or enhancing caspase-1 activity. Notably, the miR-194-5p mimic significantly enhanced the cytotoxic effects of the polytrauma cocktail, while miR-499a-5p boosted effects of LPS/nigericin stimulation in cardiomyocytes. Our findings identify miR-122-5p and miR-885-5p as potential biomarkers reflecting the cardiomyocyte response to polytrauma-induced inflammation, while miR-499a-5p and miR-194-5p appear to play a direct role in myocardial injury after polytrauma.

## 1. Introduction

Polytrauma, characterized by multiple concurrent injuries, represents a significant challenge in emergency medicine and critical care, with thoracic injury being a leading contributor to increased morbidity and mortality [1,2]. Among the various types of thoracic injuries, cardiac damage is a particularly life-threatening complication in polytraumatized patients [3,4]. Early and accurate diagnosis of myocardial injury is paramount to improve patient outcomes. The current gold standard in diagnosis is the serial measurements of troponin T (TnT), although the informative value can be limited by confounding factors, such as renal dysfunction, which can obscure the diagnosis of cardiac damage [5]. There is a need for validated biomarkers that can be used for prognosis and the detection of traumatic cardiac dysfunction.

Recent advances in molecular biology have highlighted the potential of miRNAs as novel biomarkers for cardiac injury. For example, miR-423-5p [6], miR-19a [7], miR-499 [8], and miR-208a [9] were described mostly in the context of ischemic damage due to myocardial infarction. miRNAs are small, non-coding RNA molecules that regulate gene expression post-transcriptionally, playing a crucial role in various physiological and pathological processes, including cardiac development, remodeling, regeneration, and apoptosis [10,11,12,13,14]. miRNAs in biofluids are stable, allowing them to evade degradation and maintain their presence in circulation [15,16], which enhances their potential as reliable biomarkers for traumatic damage. Polytrauma induces a rapid release of inflammatory mediators which interact with miRNA expression profiles in the circulation, thereby affecting cardiomyocyte viability and function. Given their involvement in complex biochemical interactions, miRNAs are potentially strong candidates as biomarkers for critically ill polytrauma patients [17].

While most of the existing research focuses on the involvement of miRNAs in cardiac injury due to myocardial infarction, their potential as biomarkers to monitor heart damage in polytrauma has received only limited attention thus far. In this study, we aimed to identify plasma-miRNAs associated with cardiac injury in polytrauma patients and evaluate their potential as early biomarkers of myocardial damage. Using a combination of next generation sequencing (NGS), RT-qPCR, and in vitro models, we investigated the expression patterns of five candidate miRNAs (miR-122-5p, miR-424-5p, miR-885-5p, miR-194-5p, and miR-499a-5p) in response to polytrauma. We also examined their correlation with parameters of transthoracic echocardiography and assessed their ability to influence key cellular stress pathways, including apoptosis and inflammation, in human cardiomyocyte models. The findings of this study provide insights into the role of miRNAs as biomarkers of myocardial injury in polytrauma.

## 2. Materials and Methods

A graphical summary of the study design is provided in Appendix A. This figure outlines patient cohort stratification, miRNA sequencing and validation workflows, and in vitro experiments investigating the functional role of selected miRNAs in myocardial injury models.

### 2.1. Patients’ Study

The present study was a prospective non-randomized study, conducted in a German level 1 trauma centre. We included polytrauma patients with an injury severity score (ISS) ≥ 16, a widely accepted threshold for defining severe multi-organ injury. All patients were admitted to the emergency department between 2017 and July 2024. Patient’s demographic and clinical characteristics are summarized in Appendix A.

#### 2.1.1. Ethical Approval

The study was approved by the Local Ethics Committee of the University of Frankfurt (approval ID 89/19, ID 375/14), and all procedures were conducted in accordance with the Declaration of Helsinki. Written informed consent was obtained from all patients or their legal representatives and healthy volunteers prior to their inclusion in the study. Clinical data, including cardiac-specific TnT levels (measured via highly sensitive electrochemiluminescence immunoassays (ECLIA, Roche, Rotkreuz, Switzerland)), were extracted from the patients’ digital case records and utilized for group stratification.

#### 2.1.2. Sample Collection and miRNA Isolation

In the initial phase of the study, plasma samples from 10 polytraumatized patients (ISS ≥ 16) were stratified into two groups based on their initial TnT concentration in the emergency room (ER). The high TnT group included patients with elevated TnT concentrations (>50 pg/mL, n = 5), while the low TnT group had lower concentrations (<12 pg/mL, n = 5). These samples were analysed by NGS to identify differentially expressed miRNAs.

In the subsequent validation phase, quantitative PCR (qPCR) was performed on plasma samples from a larger cohort of 19 polytraumatized patients (ISS ≥ 16), similarly stratified into high (>50 pg/mL, n = 10) and low TnT groups (<12 pg/mL, n = 9). The samples were collected at the ER for analysis.

Additionally, a third cohort consisting of 15 polytraumatized patients (ISS ≥ 16), who underwent transoesophageal echocardiography, and 7 healthy volunteers (no medication, no preexisting illnesses, and age > 18 years) were included to compare miRNA expression levels in polytraumatized patients at different time points post-trauma with those of healthy controls. This cohort also allowed for the correlation of miRNA expression data with functional outcomes obtained from transoesophageal echocardiography and relevant laboratory markers. Blood samples from patients were collected immediately upon arrival at the ER and at 24 h and 48 h post-trauma.

All samples were drawn into EDTA tubes, and plasma was separated by centrifugation at 300× *g* for 15 min at 4 °C. The plasma was aliquoted and stored at −80°C until further processing. For miRNA isolation, plasma samples (minimum 200 µL per sample) were thawed on ice, and miRNAs were extracted using the miRNeasy Serum/Plasma Advanced Kit (Qiagen, Hilden, Germany) according to the manufacturer’s instructions.

#### 2.1.3. NGS

To identify whether miRNAs are differentially expressed in the high TnT and low TnT groups, NGS and differential expression analysis were performed on the miRNAs isolated from the first cohort (n = 5 each group). Small RNA libraries were prepared by GenXPro GmbH (Frankfurt, Germany). Adapters containing molecular identifiers were attached to small RNAs before reverse transcription and PCR amplification. Sequencing was conducted on an Illumina NextSeq500 platform with 1 × 75 base pairs. Unprocessed sequencing reads underwent adapter and quality trimming using Cutadapt (version 4.6) [18]. Deduplication based on unique molecular identifiers was carried out using in-house scripts. Quality checks were performed with FastQC. Reads were then mapped to the reference genome using Bowtie2 (version 2.4.4) [19]. Transcript quantification was conducted using HTSeq (version 2.0.2) [20], and differential expression analysis was carried out using DESeq2 (version 1.38) [21]. Finally, MultiQC was used to generate a comprehensive summary report of the results.

#### 2.1.4. miRNA Expression Analysis

For RT-qPCR analysis, miRNA was extracted from 200 µL of plasma (second cohort: n = 19; third cohort: polytrauma patients, n = 16, healthy volunteers, n = 7) using the miRNeasy Serum/Plasma Advanced Kit (Qiagen, Hilden, Germany). cDNA was synthesized using the miRCURY LNA RT Kit (Qiagen, Hilden, Germany) according to the manufacturer’s instructions. To monitor cDNA synthesis efficiency, 0.5 µL of spike-ins (UniSp6 and cel-miR-39-3p) were added to each reaction. A total of 6.5 µL of RNA was used for each cDNA reaction. The cDNA was diluted 2-fold with RNase-free water, and 1 µL of the diluted cDNA was used per RT-qPCR reaction. RT-qPCR was performed using the miRCURY SYBR Green PCR Kit (Qiagen, Hilden, Germany) on a CFX96 Real-Time PCR Detection System (BioRad, Puchheim, Germany) with the following cycling conditions: an initial denaturation at 95 °C for 3 min, followed by 40 cycles of 95 °C for 10 s and 56 °C for 50 s. Commercial primers for miRNAs (cel-miR-39-3p, UniSp6, hsa-miR-122-5p, hsa-miR-424-5p, hsa-miR-194-5p, hsa-miR-499a-5p, and hsa-miR-885-5p) were purchased from Qiagen (miRCURY LNA miRNA PCR Assay, Hilden, Germany). Relative quantification was performed using the delta Ct method (2−∆Ct), with normalization to the spike-in control cel-miR-39-3p. All samples were run in duplicates and were repeated in three independent experiments.

#### 2.1.5. Transthoracic Echocardiography (TTE)

Upon admission to the intensive care unit, transthoracic echocardiography was conducted at two time points (24 h, 48 h) by an experienced cardiologist. The examination included the assessment of key parameters such as ejection fraction (EF), the E/E’ ratio (the ratio of early diastolic transmitral flow velocity to early diastolic mitral annular velocity), thickness of the interventricular septum and left ventricular posterior wall (IVS/LVPW), and tricuspid annular plane systolic excursion (TAPSE). All measurements were performed in collaboration with the Department of Cardiology at the University of Frankfurt, adhering to the general echocardiography standards set by the German Centre for Cardiovascular Research (DZHK) [22]. The echocardiographic examinations were performed by mean of Venue Go R2 and Vivid iq ultrasound systems (GE HealthCare; Düsseldorf, Germany).

### 2.2. In Vitro Study

Human cardiomyocyte (HCM) cells were cultured and treated with either polytrauma cocktail (PTC) or lipopolysaccharide (LPS) and nigericin co-stimulation to model myocardial injury. Cells were transfected with candidate miRNA mimics, and miRNA expression, cell viability, apoptosis, and inflammasome activation were assessed.

#### 2.2.1. Cell Culture and Treatment

Primary HCM cells, isolated from the ventricles of the human adult heart, were purchased from Celprogen (Catalog number: 36044-15VT, Torrance, CA, USA). Cells were cultured in myocyte growth medium (Celprogen, Torrance, CA, USA) supplemented with 10% foetal bovine serum (FBS, Gibco, Paisley, UK) at 37 °C and 5% CO_2_. Medium was refreshed every 3 days, and after reaching 70–80% confluence, cells were passaged.

For experiments, HCM cells were seeded at a density of 2 × 104 cells/cm2 in 96-well and 24-well plates and allocated into experimental (PTC—or LPS/nigericin treatment) or control (no treatment) groups. When cells reached 80–90% confluence, culture medium was replaced with freshly prepared PTC—culture medium containing 10 ng/mL activated complement factor 5 (C5a, Merck, Kenilworth, NJ, USA), 500 ng/mL activated complement factor 3 (C3a, Merck, Kenilworth, NJ, USA), 250 pg/mL interleukin 1 beta (IL-1β, Peprotech, Rocky Hill, NJ, USA), 500 pg/mL interleukin 6 (IL-6, Biomol, Hamburg, Germany), 150 pg/mL interleukin 8 (IL-8, Biomol, Hamburg, Germany), and 10 ng/mL tumour necrosis factor alpha (TNF-α, Biomol, Hamburg, Germany) and cultured for 4 h at 37 °C and 5% CO₂. The composition of the PTC was adapted from a previously published protocol [23].

In the second experimental group, the medium was replaced with freshly prepared culture medium containing 10 µg/mL LPS (Sigma-Aldrich, Taufkirchen, Germany) and incubated for 4 h. Subsequently, the medium was replaced with freshly prepared FBS-free medium containing 10µM nigericin (Cayman chemical, MI, USA), a potent activator of the NOD-like receptor protein 3 (NLRP3) inflammasome, for an additional 1-h incubation. This combined treatment is hereafter referred to as ‘‘LPS + nigericin’’. Control cells were treated with medium containing equivalent volumes of solvent, PBS.

#### 2.2.2. Cellular miRNA Isolation and miRNA Expression Analysis

For cellular miRNA isolation, experimental or control cells were washed twice with PBS without Ca^2^⁺/Mg^2^⁺ (PBS, Gibco, Paisley, UK) and lysed with 260 µL RLT lysis buffer (Qiagen, Hilden, Germany), and miRNAs were isolated via the miRNeasy Tissue/Cell Kit (Qiagen, Hilden, Germany) following the manufacturer’ protocol. cDNA synthesis and RT-qPCR for miRNA expression analysis were performed as described before. Relative quantification was performed using the delta–delta Ct method (2-ΔΔCt) with normalization to the spike-in control UniSp6 or cel-miR-39-3p. All samples were run in duplicate and were repeated in three independent experiments.

#### 2.2.3. miRNA Transfection

To transiently elevate the expression of target miRNA in HCM cells, miR-122-5p (GeneGlobe ID: YM00470430), miR-424-5p (GeneGlobe ID: YM00472282), miR-885-5p (GeneGlobe ID: YM00470666), miR-194-5p (GeneGlobe ID: YM00471895), and miR-499a-5p (GeneGlobe ID: YM00470181) mimics were purchased from Qiagen (Qiagen, Hilden, Germany). A FAM-labeled miRCURY LNA miRNA mimic negative control (GeneGlobe ID: YM00479902, Qiagen, Hilden, Germany) was used as a negative control. HCM cells were seeded at a density of 2 × 104 cells/cm2 in 96-well and 24-well plates and cultured in myocyte growth medium for 24 h.

Cells were then transfected with 5 nM each of miRNA mimic using the Lipofectamine RNAiMAX transfection reagent (Thermo Fisher Scientific, Carlsbad, CA, USA).

The miRNA mimics were first diluted to 10 nM, and the Lipofectamine RNAiMAX was diluted to 4 µL/mL in myocyte growth medium without FBS. Equal volumes of the diluted miRNA mimics and the transfection reagent were mixed and incubated at room temperature for 5 min to allow for complex formation. Then 100 µL and 500 µL of medium containing the complex were added to the cells in 96-well plates and 24-well plates, respectively. After 4 h of incubation, the medium was replaced with fresh myocyte growth medium containing 10% FBS, and cells were incubated for an additional 24 h. All reactions were performed in duplicate. Transfection efficiency of miRNA mimic was analysed using qPCR (Appendix A).

#### 2.2.4. Cell Viability Assay

To assess the potential cytotoxic effects of PTC, LPS + nigericin, and miRNA mimics on HCM cells, cell viability was evaluated across control, blank, and experimental groups using the alamarBlue reagent (Bio-Rad, Puchheim, Germany) according to the manufacturer’ protocol. After treatment, cells were incubated with fresh medium containing alamarBlue (10% (*v*/*v*)) for 4 h at 37 °C and 5% CO_2_. The absorbance of the conditioned medium was then measured at excitation 540 nm and emission 590 nm using a microplate reader (Tecan Infinite 200, Tecan, Crailsheim, Germany). The percentage reduction of alamarBlue was calculated following the manufacturer’s guidelines, and data were normalized by setting the mean % reduction of the control group to 1. All samples were analysed in triplicate, and three independent experiments were performed. Data were presented as mean values with standard deviations (SD).

#### 2.2.5. Caspase-1 Activity Assay

To assess caspase-1 activity in HCM cells after miRNA mimics transfection, with or without additional treatment (including PTC or LPS + nigericin), the conditioned medium was collected and incubated with the luciferin WEHD-substrate from the Caspase-Glo 1 Assay kit (Promega, Madison, WI, USA). The Caspase-Glo 1 reagent was prepared according to the manufacturer’s instructions and added to the samples at a 1:1 ratio. After 1 h incubation at room temperature in the dark, luminescence was measured using a microplate reader (Tecan Infinite 200, Tecan, Crailsheim, Germany). All samples were analysed in triplicate, and three independent experiments were performed. Data are presented as fold changes relative to the control group.

#### 2.2.6. Apoptosis Assay

Apoptosis in HCM cells was assessed using the Caspase–Glo^®^ 3/7 assay according to the manufacturer’s instructions (Promega, Madison, WI, USA). Briefly, HCM cells were seeded in a 96-well plate at a density of 15,000 cells/well and cultured for 24 h prior to transfection with miRNA mimics, with or without additional treatment, including PTC or LPS + nigericin stimulation. Following treatment, 100 μL of reconstituted caspase–Glo^®^ 3/7 reagent was added to each well. Cells were then incubated at room temperature for 1 h in the dark, and luminescence was measured using a microplate reader (Tecan Infinite 200, Tecan, Crailsheim, Germany). All samples were analysed in triplicate, and three independent experiments were performed. Data are presented as fold changes relative to the control group.

#### 2.2.7. Statistical Analysis

For the NGS-based differential expression analysis, DESeq2 was used to identify differentially expressed miRNAs between the high TnT and low TnT groups. To correct for multiple testing, the Benjamini–Hochberg (BH) false discovery rate (FDR) method was applied, ensuring robust control of false positives in high-throughput sequencing data.

Other statistical analyses were performed using GraphPad Prism version 9 software (GraphPad Software Inc., La Jolla, CA, USA). The Kolmogorov–Smirnov test was used to assess the normality of data distribution. For normally distributed data, an unpaired *t*-test was used to assess statistical significance (*p* < 0.05) between two groups. For not normally distributed data, the Mann–Whitney U test was applied. For comparisons involving more than two groups, one-way analysis of variance (ANOVA) with Dunnett’s test was performed, followed by t-tests where necessary (e.g., for comparison between the LPS + nigericin group and miR-499a-5p mimic transfection combined with the LPS + nigericin group (Figure 4); for comparison between the PTC group and miR-194-5p mimic transfection combined with the PTC group, and between LPS + nigericin stimulation group and miR-194-5p mimic transfection combined with LPS + nigericin stimulation group (Figure 5). Correlation analysis was carried out with Spearman’s rank correlation. Data are presented as the mean ± standard deviation (SD). A *p*-value < 0.05 was considered statistically significant, and a *p*-value > 0.05 and <0.1 was rated as statistical trend.

## 3. Results

### 3.1. Plasma miRNA Profiling in Polytrauma Patients with Different TnT Levels

#### 3.1.1. Identification of Differentially Expressed miRNAs Associated with Cardiac Injury in Polytrauma Patients

A total of 10 patients with polytrauma, defined by an ISS ≥ 16, were enrolled and stratified into two groups: those with cardiac injury (TnT > 50 pg/mL) and those without cardiac injury (TnT < 12 pg/mL). Plasma samples were collected immediately at the ER, and miRNAs were subsequently isolated. Next-generation sequencing (NGS) was then performed to profile the miRNA expression, revealing 57 differentially expressed miRNAs between the two groups (GEO accession number: GSE286533). Following this, we applied a filtering criterion of fold change (|fold change| > 1.8, *p* < 0.05) and consulted the HMDD v4.0 database [24] to assess the association of these miRNAs with cardiovascular disease, ultimately identifying ten miRNAs of interest (Table 1).

#### 3.1.2. Validation of the Selected Differentially Expressed miRNAs in Polytrauma Patients

From the initial analysis, ten miRNAs of interest were identified based on significant differential expression between polytrauma patients with high and low TnT levels and their potential association with cardiovascular disease. Given that miR-122-5p was significantly elevated in patients with cardiac injury, indicating its potential role in myocardial damage [25,26], we also included this miRNA in further analysis.

To validate the differential expression of the selected miRNAs, quantitative RT-qPCR was performed in a larger cohort of 19 patients. Among the candidates, five miRNAs—miR-122-5p (*p* < 0.01), miR-424-5p (*p* < 0.05), miR-885-5p (*p* < 0.01), miR-194-5p (*p* < 0.05), and miR-499a-5p (*p* < 0.05)—exhibited statistically significant differences in expression between the high TnT and low TnT groups (Figure 1). Specifically, these five miRNAs showed higher expression in the group with elevated TnT levels compared to the low TnT group (Figure 1).

#### 3.1.3. Elevated Expression of Candidate miRNAs as Early Predictors of Cardiac Injury in Polytrauma Patients

To investigate the potential of the selected miRNAs as early predictors of cardiac injury following polytrauma, plasma samples were collected from an independent cohort of 15 polytrauma patients at three different time points (at the ER and 24 h and 48 h post-trauma) and healthy volunteers (n = 7) as controls. Expression analysis revealed that the expression levels of the five candidate miRNAs (miR-122-5p, miR-885-5p, miR-424-5p, miR-194-5p, and miR-499a-5p) were significantly higher at the ER time point compared to both the later time points and to expression levels of healthy control group (Figure 2).

In parallel, patients in this cohort underwent TTE assessment to monitor cardiac function, including parameters such as the E/E’ ratio, EF, TAPSE, and IVS/LVPW. These echocardiographic parameters were correlated with the expression levels of the candidate miRNAs to investigate potential associations between miRNA expression and myocardial damage following polytrauma. The results of Spearman’s correlation analysis indicated that expression of miR-885-5p at the ER was strongly correlated with the development of diastolic dysfunction, as measured by the E/E’ ratio at 24 h (r = 0.93, *p* < 0.05, Table 2). Additionally, miR-122-5p at the ER showed a negative correlation with TAPSE at 48 h (r = −0.89, *p* < 0.01, Table 2), suggesting an association with right ventricular dysfunction. miR-499a-5p at the ER correlated negatively with EF at 24h (r = −0.63, *p* = 0.1, Table 2), indicating its potential role in impaired left ventricular function, though this correlation did not reach statistical significance. Moreover, miR-885-5p (r = 0.97, *p* < 0.001, Table 2) and miR-122-5p (r = 0.85, *p* < 0.05, Table 2) at the ER correlated positively with left ventricular hypertrophy, as measured by IVS/LVPW at 24 h. On the other hand, miR-194-5p also demonstrated a positive correlation with IVS/LVPW at 24 h, though this correlation did not reach statistical significance (r = 0.67, *p* = 0.08, Table 2).

The results highlight significant correlations that reinforce the potential role of these miRNAs in assessing cardiac function post-trauma.

### 3.2. Functional Analysis of Candidate miRNAs in HCM Cells

#### 3.2.1. miRNA Expression Pattern in the Two Myocardial Injury Cell Models

To explore the potential roles of differentially expressed miRNAs in post-traumatic cardiac damage, two in vitro cardiomyocytes injury models were established with HCM cells. In the first model, HCM cells were treated with PTC [23] for 4 h. In the second model, HCM cells were primed with LPS (10 µg/mL) for 4 h, followed by stimulation with nigericin (10 µM) for 1 h in serum-free condition (LPS + nigericin).

Both PTC and LPS + nigericin stimulation significantly reduced cell viability, promoted apoptosis, and upregulated the expression of inflammasome-related genes (Appendix A). Notably, PTC stimulation enhanced caspase-1 activity, indicating an activation of the inflammasome pathway. In contrast, LPS + nigericin stimulation did not result in a measurable increase in caspase-1 activity (Appendix A). These findings suggest that while both treatments induce cellular stress and apoptosis, the mechanisms of inflammasome activation differ between the two models, with PTC exhibiting a more pronounced effect on caspase-1 activation.

Following the initial analysis of cell viability, we investigated the effects of the PTC and LPS + nigericin treatments on the expression of the selected miRNAs in HCM. The results showed that the expression of miR-885-5p and miR-122-5p was upregulated in both models. Specifically, miR-885-5p (*p* < 0.05) was significantly upregulated in both models, while miR-122-5p (*p* = 0.07) was also upregulated but did not reach statistical significance (Figure 3). The expression levels of the remaining miRNAs, miR-194-5p, miR-424-5p, and miR-499a-5p, showed no significant changes in either model (Figure 3).

#### 3.2.2. Impact of Candidate miRNA Mimics on HCM Cell Viability in Two Myocardial Injury Cell Models

To assess the functional impact of each candidate miRNA on HCM cells, the cells were individually transfected with mimics for miR-122-5p, miR-424-5p, miR-885-5p, miR-194-5p, and miR-499a-5p for 4 h, respectively. Cell viability was then assessed using the AlamarBlue assay. Among the five miRNAs tested, miR-194-5p and miR-499a-5p mimic transfection significantly reduced HCM cell viability (*p* < 0.05, Figure 4A).

To investigate the effects of these miRNAs in the PTC- injury HCM model, cells were individually transfected with each miRNA mimic and subsequently treated with PTC for 4 h. The results of the cell viability analysis showed that both, PTC treatment alone or in combination with miRNA mimic transfection, significantly reduced the cell viability (Figure 4B). Notably, in the case of the miR-194-5p mimic, the combinatoric treatment (mimic transfection and PTC) has a significantly stronger effect on cell viability than the PTC treatment alone (*p* < 0.05, Figure 4B).

In similar experiments performed in the LPS + nigericin injury model, LPS + nigericin treatment, both alone and in combination with the miRNA mimic transfection, was found to have a significant negative effect on HCM viability (Figure 4C). Among the investigated miRNAs mimics, a cumulative negative effect of the LPS + nigericin treatment and the mimic transfection was found for miR-499a-5p mimic (*p* < 0.05, Figure 4C).

**Figure 4 cells-14-00300-f004:**
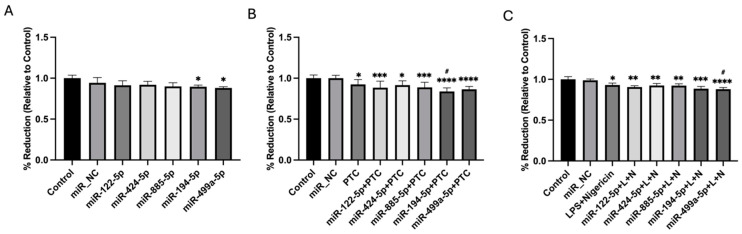
Impact of selected miRNA mimics, with and without PTC or LPS + nigericin treatment, on viability of HCM cells. (**A**) HCM cells were transfected individually with each selected miRNA mimic, and cell viability was assessed using the assay in control (non-treated), miR_NC (miRNA negative control) and selected miRNA-transfected cells. (**B**) Following a 4-h transfection with selected miRNA mimics, HCM cells were incubated for another 24 h and then treated with PTC for 4 h. Cell viability was then assessed using the alamarBlue assay in control (non-treated), miR_NC (negative control) and selected miRNA-transfected cells. (**C**) After a 4 h-transfection with selected miRNA mimics, cells were incubated for 24 h and then treated with LPS (10 µg/mL, 4 h) and nigericin (10 µM, 1 h). Cell viability was measured using the alamarBlue assay in control (non-treated), miR_NC (negative control) and cells transfected with selected miRNA mimics. * *p* < 0.05, ** *p* < 0.01, *** *p* < 0.001, **** *p* < 0.0001 represent comparison of transfected group vs. control group. # *p* < 0.05 represents comparison of transfected group vs. PTC group (**B**) or transfected group vs. LPS + nigericin group (**C**).

#### 3.2.3. Influence of Candidate miRNAs on Apoptosis and Inflammasome Activation in Two Myocardial Injury Cell Models

After showing effects of selected miRNA mimics on HCM cell viability, we further investigated their influence on apoptosis and caspase-1 activation, the key processes involved in myocardial injury, using the same in vitro myocardial injury models.

First, we evaluated the effects of individual miRNA mimics on caspase 3/7 activity (apoptosis). Both the miR-194-5p mimic (*p* < 0.0001) and the miR-499a-5p mimic (*p* < 0.05) significantly increased caspase 3/7 activity (Figure 5A). Next, the combinatoric effect of miRNA mimics and PTC treatment was evaluated. The miR-194-5p mimic showed a significantly greater increase in caspase 3/7 activity compared to PTC treatment alone (*p* < 0.0001, Figure 5B). In case of LPS + nigericin treatment, no additional effect of any miRNA mimic transfections was observed (Figure 5C).

In addition, the effects of miRNA mimics on caspase-1 activity were investigated in our in vitro models. We found that out of five selected miRNAs, only miR-499a-5p has a significant effect on caspase-1 activity (*p* < 0.001, Figure 5D). PTC treatment alone significantly increased caspase-1 activity (*p* < 0.0001), and the combination of PTC with transfection of miR-122-5p (*p* < 0.01), miR-885-5p (*p* < 0.001), miR-194-5p (*p* < 0.0001), or miR-499a-5p (*p* < 0.001) mimics further enhanced the increase in caspase-1 activity (Figure 5E). In contrast, LPS + nigericin alone did not significantly affect caspase-1 activity. Notably, a combinatory effect of LPS + nigericin and mimic transfection on caspase-1 activity was observed only for miR-499a-5p (*p* < 0.001, Figure 5F).

**Figure 5 cells-14-00300-f005:**
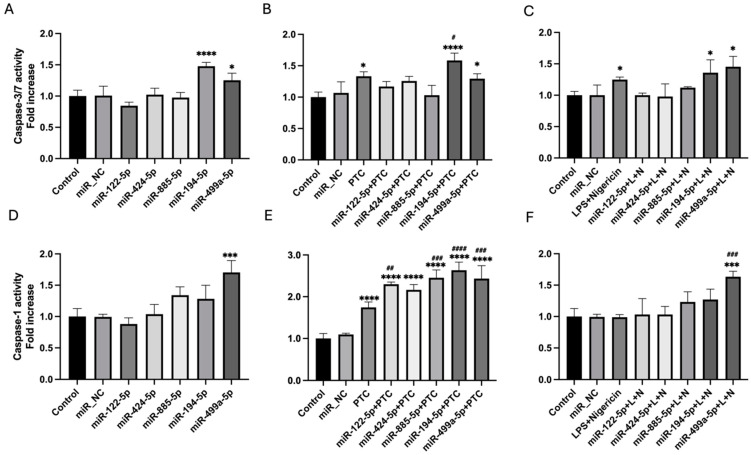
Impact of miRNA mimics (with or without PTC or LPS + nigericin treatment) on apoptosis and inflammasome activity in HCM cells. (**A**) HCM cells were individually transfected with each selected miRNA mimic, and cell apoptosis was assessed by Caspase 3/7 activity assay in control (non-treated), miR_NC (miRNA negative control), and miRNA-transfected cells. (**B**) Following a 4-h transfection with selected miRNA mimics, cells were incubated for 24 h and then treated with PTC for 4 h. HCM apoptosis was quantified by Caspase-3/7 activity assay. (**C**) After a 4-h transfection with selected miRNA mimics HCM cells were incubated for 24 h and then treated with LPS (10 µg/mL, 4 h) and nigericin (10 µM, 1 h). HCM apoptosis was measured by Caspase-3/7 activity assay. (**D**) HCM cells were individually transfected with each miRNA mimic, and caspase-1 activity was assessed using the Caspase-Glo 1 Assay kit (**E**) Following a 4-h transfection with selected miRNA mimics, cells were incubated for 24 h and then treated with PTC for 4 h. Caspase-1 activity was measured using the Caspase-Glo 1 Assay kit. (**F**) After a 4-h transfection with selected miRNA mimics HCM cells were incubated for 24 h and then treated with LPS (10 µg/mL, 4 h) and nigericin (10 µM, 1 h). Caspase-1 activity was assessed using the Caspase-Glo 1 Assay kit. * *p* < 0.05, *** *p* < 0.001, **** *p* < 0.0001 represent comparison of transfected group vs. control group. # *p* < 0.05, ## *p* < 0.01, ### *p* < 0.001, #### *p* < 0.0001 represent comparison of transfected group vs. PTC group (**B**,**E**) or transfected group vs. LPS + nigericin group (**F**).

## 4. Discussion

### 4.1. Cardiac Injury Caused by Polytrauma Is Reflected in Plasma miRNA Profiles

miRNAs have emerged as critical regulators of cardiac injury following polytrauma and show high potential as diagnostic biomarkers and therapeutic targets. In this study, we hypothesized that cardiac injury caused by polytrauma could be reflected in plasma miRNA profiles and aimed to identify plasma miRNAs that are associated with cardiac injury in polytrauma patients. Therefore, we compared miRNA profiles among polytrauma patients with and without elevated TnT levels, the gold standard of myocardial injury, by mean of NGS and RT-qPCR. We found five miRNAs (miR-122-5p, miR-424-5p, miR-885-5p, miR-194-5p, and miR-499a-5p) to be significantly upregulated in patients with elevated TnT levels. Moreover, our findings in a larger cohort of patients and healthy controls showed that these miRNAs were significantly upregulated specifically at the ER in patients with elevated TnT level. These results supported our hypothesis and suggested the potential involvement of these miRNAs in early physiological response to trauma and cardiac injury. This indicates a potential use as biomarker.

To further demonstrate a possible connection between these miRNAs and cardiac injury, we correlated their expression levels at the ER with cardiac dysfunction measured via transthoracic echocardiography in our patients. The results of this analysis showed that the expression of these miRNAs likely reflected impairments in cardiac function. Thus, the development of diastolic dysfunction (measured by the E/E’ ratio at 24 h) was reflected by an increase in miR-885-5p expression. The dysfunction of the right ventricular (TAPSE at 48 h) was associated with miR-122-5p upregulation, whereas dysfunction of the left ventricular (EF at 24h) was associated with miR-499a-5p. In addition, a possible connection between left ventricular hypertrophy (IVS/LVPW at 24 h) and the miR-885-5p, miR-122-5p and miR-194-5p was observed. These findings further support the potential use of these miRNAs as biomarkers of cardiac injury post-trauma.

### 4.2. The Potential Role of Polytrauma-Induced miRNAs in Cardiomyocytes Injury

While it is impossible to fully replicate the complexity of polytrauma in vitro, we aimed to mimic its major inflammatory aspects to study cardiomyocyte injury. We hypothesized that elevated plasma levels of these miRNAs could either reflect or promote the cardiomyocyte damage resulting from polytrauma. To explore these possibilities, we conducted an in vitro study using two human cardiomyocyte injury models. In one of these models, the unique inflammatory milieu of polytrauma was mimicked by adding PTC [23] to the cell culture medium, whereas in the other inflammasome activation [27,28,29] was induced via LPS and nigericin treatment.

The results of the in vitro study showed that polytrauma injury (mimicked by PTC or LPS + nigericin treatment) significantly increased the expression of miR-885-5p and miR-122-5p, but not of the other miRNAs, in cardiomyocytes. These results were in accordance with previously published data, showing upregulation of miR-122-5p while cardiac damage [30,31], and elevated miR-885-5p in human myocardial cells treated with exosomes isolated from sepsis patients [32]. These findings suggested that increased levels of miR-885-5p and miR-122-5p in the plasma of polytrauma patients could likely reflect the damaged cardiomyocytes. Together with the observed correlation between the miRNA expression and cardiac dysfunction in patients, these results highlight the potential of these miRNAs as early cardiac injury biomarkers in polytrauma patients. However, further validation studies are needed to confirm these findings and investigate their clinical relevance.

To explore whether high levels of target miRNAs could negatively affect cardiomyocytes (and therefore promote/enhance cardiac injury), we evaluated their effects through transfection with miRNA mimics in our in vitro models. Our results showed that the miR-194-5p mimic (alone) and the miR-499a-5p mimic (alone or in combination with PTC) had a significant negative effect on HCM viability via enhancement of apoptosis (upregulation of caspase 3/7) and, in the case of the miR-499a-5p, via inflammasome activation (increased caspase-1 activity). These results were consistent with previous studies, in which miR-194-5p, upregulated in the hearts of septic mice, was correlated with enhanced myocardial injury [33], and upregulated in plasma miR-499a-5p correlated with hypertrophic cardiomyopathy [34]. Moreover, it was previously shown that the miR-194-5p exacerbates cardiac dysfunction, inflammation, oxidative stress, apoptosis, and mitochondrial dysfunction in septic mice [33]. Additionally, upregulation of miR-194-5p was shown to compromise the antiapoptotic function of X-box binding protein 1 (XBP1s) in cardiomyocytes [35]. However, there are studies suggesting that miR-194-5p may have protective effects on cardiomyocytes under stress. Specifically, miR-194-5p overexpression was found to protect against myocardial injury due ischaemia/reperfusion by inhibiting apoptosis and oxidative stress via mitogen-activated protein kinase 1 (MAPK1) [36]. The role of miR-499a-5p also appeared to be dual. While other studies have reported downregulation of the miR-499a-5p in myocardial ischaemia/reperfusion (I/R) injury [37] and its protective role against H_2_O_2_-induced cardiomyocyte apoptosis [38], our present findings suggested that miR-499a-5p may play a cytotoxic role in polytrauma-induced cardiac injury. These contradictory findings could be explained by a context-dependent role of both miRNAs in cardiac injury. These findings support the hypothesis that elevated plasma levels of miR-194-5p and miR-499a-5p may serve as prognostic markers for adverse outcomes in cardiac injury. However, further studies are needed to clarify their precise role and potential as clinical biomarkers in polytrauma-induced cardiac damage.

Regarding miR-424-5p, we did not find any significant effects of its mimic on HCM cell viability, apoptosis, or caspase-1 activity. Other studies have shown that miR-424-5p is significantly downregulated in heart failure [39], while is upregulated in patients with fatal myocardial infarction [40], suggesting that the role of this miRNA in cardiac pathology may involve more complex regulatory mechanisms. One possible explanation of the observed absence of the effects could be the relatively high endogenous expression level of this miRNA in HCM cells. Such elevated endogenous expression may result in a saturation effect, where further exogenous transfection of miR-424-5p mimic fails to produce additional biological effects. The miRNA saturation effects have been described previously, and high levels of endogenous miRNAs were shown to compete with exogenous miRNA (mimics) for incorporation into the RNA-induced silencing complex, therefore reducing the efficacy of the mimics [41]. Further research should explore the precise mechanisms of miR-424-5p in cardiac injury, including pathways beyond those assessed in the present study.

While our study provides mechanistic insights into miRNA-mediated cardiomyocyte injury, in vitro models cannot fully replicate the complexity of polytrauma, particularly the multifactorial inflammation and multi-organ interactions involved. Although miR-499a-5p is highly expressed in cardiomyocytes [9,42,43,44], the other identified miRNAs have been detected in non-cardiac tissues [45,46]. A study demonstrates that hepatic injury leads to the release of miR-122 into circulation, where it can mediate systemic inflammatory responses, including pulmonary inflammation [47]. Additionally, miR-885-5p has been shown to be significantly downregulated in patients with intracranial haemorrhage secondary to traumatic brain injury, and its reduced levels correlate with increased neuroinflammation and worse clinical outcomes [48]. These findings suggest that circulating miRNAs may originate from multiple organs affected by polytrauma, rather than exclusively from the heart. While their correlation with TnT levels and echocardiographic parameters suggests a link to myocardial injury, their precise tissue origin in the context of polytrauma remains to be clarified. Future studies should employ tissue-specific analyses and extracellular vesicle profiling to further define the relative contributions of cardiac and non-cardiac sources.

Circulating miRNAs can be released through multiple mechanisms, including active secretion via extracellular vesicles (EVs), passive leakage from necrotic cells, and apoptotic bodies [43,49]. The distinction between miRNAs that are actively secreted as signalling molecules and those passively released due to cellular damage is crucial in understanding their functional role in myocardial injury. Given the complexity of miRNA secretion dynamics potential multi-organ contributions, additional studies utilizing tissue-specific expression analyses and EV profiling are warranted to better define their specificity for cardiac damage.

While our study identified potential biomarkers (miR-122-5p, miR-885-5p, miR-499a-5p, and miR-194-5p), the observed correlations with cardiac dysfunction need to be confirmed in larger patient cohorts and with longer follow-up periods. Additionally, validating the association between miRNA expression and echocardiographic parameters in a larger patient collective is essential to account for potential confounding factors such as age and pre-existing conditions. Future research should also investigate the target genes and pathways regulated by these miRNAs, employing miRNA inhibitors to further elucidate their role and therapeutic potential in myocardial injury. Standardizing miRNA detection protocols and integrating their profiles with established diagnostic tools could pave the way for personalized approaches to diagnosing and treating cardiac injury in polytrauma patients.

## 5. Conclusions

This study showed that cardiac injury resulting from polytrauma is reflected in plasma miRNA profiles. We identified five miRNAs (miR-122-5p, miR-424-5p, miR-885-5p, miR-194-5p, and miR-499a-5p) as significantly upregulated in polytrauma patients with elevated TnT levels, highlighting their potential as early indicators of myocardial damage. Among these, miR-122-5p and miR-885-5p were found to reflect cardiomyocytes response to the injury signalling, which further supported their potential as diagnostic biomarkers, whereas miR-499a-5p and miR-194-5p were found to play more active roles in cardiomyocyte injury, promoting apoptosis and inflammasome activation. Given their strong association with myocardial damage and dysfunction in polytrauma patients, these miRNAs may serve as indicative biomarkers for cardiac injury. Future studies should validate these findings in larger patient cohorts, elucidate the mechanisms underlying these miRNAs in cardiac injury, and assess their prognostic and therapeutic potential in clinical settings.

## Figures and Tables

**Figure 1 cells-14-00300-f001:**
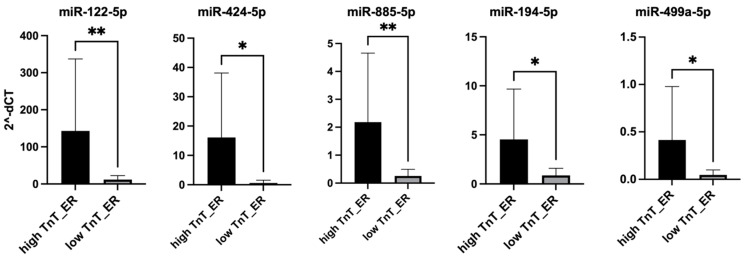
Validation of the differential expression of selected miRNAs in a larger cohort of polytrauma patients (n = 19). The expression levels of miR-122-5p, miR-885-5p, miR-424-5p, miR-194-5p, and miR-499a-5p were assessed using RT-qPCR analysis in polytrauma patients with low (low TnT_ER, TnT < 12 pg/mL) and high (high TnT_ER, TnT > 50 pg/mL) TnT concentration at the ER. * *p* < 0.05, ** *p* < 0.01. TnT: Troponin T; ER: emergency room.

**Figure 2 cells-14-00300-f002:**
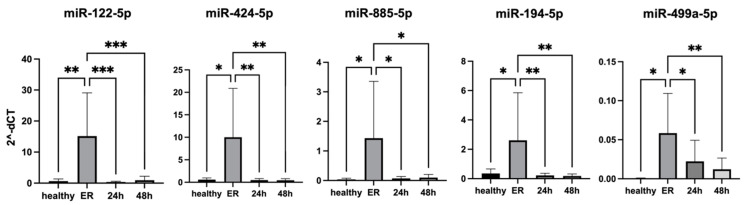
Candidate miRNA expression profiles in polytrauma patients at different time points. Plasma samples were collected from polytrauma patients (n = 15) at three different time points: at the emergency room (ER) and 24 h and 48 h post-trauma. Additionally, plasma samples from healthy volunteers (n = 7) were collected as controls. Expression levels of selected miRNAs were assessed using RT-qPCR. * *p* < 0.05, ** *p* < 0.01, *** *p* < 0.001.

**Figure 3 cells-14-00300-f003:**
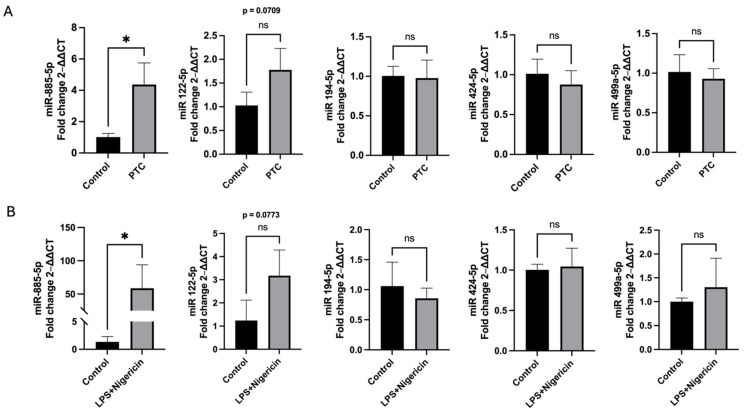
Both polytrauma cocktail and LPS + nigericin stimulations lead to an upregulation of miR-885-5p and miR-122-5p expression in HCM cells. (**A**) HCM cells were treated with a polytrauma cocktail (PTC) containing C3a, C5a, IL-1β, IL-6, IL-8, and TNF-α for 4 h. miRNAs were isolated, and RT-qPCR was performed to assess the expression levels of miR-885-5p, miR-122-5p, miR-194-5p, miR-424-5p, and miR-499a-5p in non-treated (control) or treated HCM cells. (**B**) HCM cells were stimulated with LPS for 4 h, followed by nigericin stimulation in serum-free medium for 1 h (LPS + nigericin). miRNA was isolated, and RT-qPCR was used to evaluate the expression levels of miR-885-5p, miR-122-5p, miR-194-5p, miR-424-5p, and miR-499a-5p in non-treated (control) or treated HCM cells. * *p* < 0.05, ns = no significance.

**Table 1 cells-14-00300-t001:** Differentially expressed miRNAs between high TnT group and low TnT group identified using NGS. FC: fold change; TnT: troponin T.

miRNA	log2 FC (Low TnT versus High TnT)	*p* Value (Low TnT versus High TnT)
hsa-miR-215-5p	−3.22	0.00
hsa-miR-618	−3.85	0.01
hsa-miR-378d	−3.93	0.04
hsa-miR-885-5p	−2.02	0.01
hsa-miR-184	−4.92	0.00
hsa-miR-424-5p	−2.26	0.00
hsa-miR-148a-5p	−1.81	0.04
hsa-miR-122b-3p	−2.37	0.04
hsa-miR-194-5p	−2.57	0.00
hsa-miR-499a-5p	−2.03	0.01

**Table 2 cells-14-00300-t002:** Correlation between the expression levels of candidate miRNAs and TTE parameters. TTE: transthoracic echocardiographic, E/E’: ratio of early diastolic transmitral flow velocity to early diastolic mitral annular velocity, EF: ejection fraction, TAPSE: tricuspid annular plane systolic excursion, IVS/LVPW: thickness of the interventricular septum and posterior wall of the left ventricle.

TTE Parameter	Candidate miRNAs	Spearman *r*	*p*-Value
E/E’_24 h	hsa-miR-885-5p	0.93	0.02
E/E’_24 h	hsa-miR-122-5p	0.81	0.07
EF_24 h	hsa-miR-499a-5p	−0.63	0.1
TAPSE_48 h	hsa-miR-122-5p	−0.89	0.01
IVS/LVPW_24 h	hsa-miR-885-5p	0.97	0.001
IVS/LVPW_24 h	hsa-miR-122-5p	0.85	0.049
IVS/LVPW_24 h	hsa-miR-194-5p	0.67	0.08

## Data Availability

The raw and processed miRNA expression data generated during this study have been deposited in the NCBI Gene Expression Omnibus (GEO) under accession number GSE286533. These data are publicly available at https://www.ncbi.nlm.nih.gov/geo/query/acc.cgi?acc=GSE286533 (accessed on 17 February 2025).

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
