# Peer review of "Dual Roles of Plasma miRNAs in Myocardial Injuries After Polytrauma: miR-122-5p and miR-885-5p Reflect Inflammatory Response, While miR-499a-5p and miR-194-5p Contribute to Cardiomyocyte Damage"

_cells, 2025, doi:10.3390/cells14040300_

Round 1
Reviewer 1 Report
Comments and Suggestions for Authors
Authors showed that five miRNAs (miR-122-5p, miR-424-5p, miR-885- 538
5p, miR-194-5p, and miR-499a-5p) were significantly upregulated in polytrauma patients with elevated TnT levels and concluded that these miRNAs were potential early indicators of myocardial damage. It is a valuable conclusion of this study. However, relationship between findings obtained from clinical samples and in vitro study is not conclusive, since they did not show the effect of the same set of miRNAs on the other lineage cells by using own data or discussion using data from literatures. This point is serious lack in this study.
Major comments:
1. Cardiac damage was evaluated with serum levels of TnT. Ten miRNAs were selected as cardiac damage-related miRNAs with significant difference from low TnT group. In addition, the high levels of selected miRNAs were observed at early stage of polytrauma. It is very worthy results in this study. However, the authors described the influence of renal damage on the serum level of TnT in general. Therefore, they should show the renal damage in each polytrauma patients. All samples were taken from patients without renal damage?
2. Results from in vitro study using human cardiomyocyte (HCM) with addition of stimulator of inflammatory agents or miRNA mimics clearly supported in vivo observation. Are these effects specific to only cardiomyocyte damage? Circulating miRNAs come from all other damaged organs other than cardiomyocytes. There are no data of effects on the other lineages of organs or cells. It is very critical point of this study since patients have multi-organ damages from polytrauma. I afraid of strong bias on cardiac damage.
3. TnT is contained in cardiomyocyte, so that observed high levels of serum TnT were results of release of TnT from damaged cardiomyocytes. miRNAs are usually ubiquitous in multi cell lineages, not specific to limited lineages. There is a lack of discussion on relationship between circulating and cellular miRNAs in polytrauma patients. Please discuss about this under pathophysiology of multi-organ damage.
Minor comments:
1. Precise definition of “polytrauma” is not mentioned in description of patients.
2. Please show reasons why 10 miRNAs for RT-qPCR assay were selected. Not all of them appeared in NGS results.
3. I wonder some confusion of miRNA name: In Table 1, hsa-miR122b-3p; in Line 283, miR-122-5p; all has different naming.
4. Figure 1 shows trans-sectional comparison of 5 miRNAs in polytrauma patients. Y-axis shows the relative levels of each miRNAs after normalization with spiked-in cel-miR-39-3p. Figure 1 should contain the levels of healthy controls.
5. In table 1, explain what “FC” means. No abbreviation I see.
6. Lines 512-521: Are there any inhibitory experiments with agents against miR424-5p such as decoy, sponge or antisense nucleic acids in your data or in previous reports? This point may be informative to discussion in this section.
Reviewer 2 Report
Comments and Suggestions for Authors
The lack of a healthy control group for direct comparisons in key analyses weakens the argument that these miRNAs are specific to trauma-induced cardiac injury rather than general inflammatory responses.
The correlation analyses between miRNA expression and echocardiographic parameters (e.g., TAPSE, IVS/LVPW) suggest potential biomarker utility, but adjustments for confounding variables (age, pre-existing conditions, ISS score) should be performed to strengthen these claims.
Moreover, in statistical analyses you should address multiple testing correction, as false positives could arise in high-throughput sequencing data.
The sensitivity and specificity of miR-122-5p and miR-885-5p, which serve as early biomarkers of cardiac injury, is not quantitatively assessed compared to Troponin T (TnT). This need further investigation.
A summary figure outlining the study design (e.g., patient selection, miRNA profiling, functional validation) would improve clarity in methods.
Reviewer 3 Report
Comments and Suggestions for Authors
Review of the manuscript ” Dual roles of plasma miRNAs in myocardial injuries after polytrauma: miR-122-5p and miR-885-5p reflect inflammatory response, while miR-499a-5p and miR-194-5p contribute to cardiomyocyte damage” for the Cells journal.
The goal of this study was to identify plasma miRNAs associated with cardiac injury in polytrauma patients and evaluate their potential as early biomarkers of myocardial damage. Authors have investigated, using a combination of Next Generation Sequencing (NGS), RT-qPCR and in vitro models, the expression patterns of five candidate miRNAs (miR-122-5p, miR-424-5p, miR-885-5p, miR-194-5p, and miR-499a-5p) in response to polytrauma. Authors also examined their correlation with parameters of transthoracic echocardiography and assessed their ability to influence key cellular stress pathways, including apoptosis and inflammation, in human cardiomyocyte models. The findings of this study provide insights into the role of miRNAs as biomarkers of myocardial injury in polytrauma.
However, during the review of this manuscript though, some remarks and comments appeared.
Minor comments:
1. There are some spelling errors throughout the manuscript that should be carefully corrected during revision.
2. Table with the clinical characteristics of the patients could give valuable information.
3. Synthetic RNA spike-ins should not be used for normalization (e.g. cel-miR-39-3p).
Reviewer 4 Report
Comments and Suggestions for Authors
The paper by Jiaoyan Han et al. (manuscript cells-3455144) attempts to identify miRNA markers that may be useful to formulate a prognosis and monitor the evolution of thoracic trauma patients with myocardial injury. A prospective clinical study explored plasma miRNA levels in polytrauma patients with high injury severity score (ISS ≥16) divided in two clinical groups (with/without myocardial injury) based on cardiac troponin T levels at presentation in the emergency room (cTnT>50 pg/ml vs. cTnT<12 pg/ml). A subsequent in-depth study on larger groups of patients explored miRNA levels at different time points: presentation at ER, 24 h, 48 h, including an additional group of polytrauma patients ) who underwent transesophageal echocardiography and a control group of healthy volunteers. By Illumina sequencing 10 differentially expressed miRNA species were identified, and 5 of them were confirmed by qRT-PCR in the larger study. The clinical study was supplemented with an in vitro study on a commercial preparation of dissociated human adult ventricular cardiomyocytes that attempted to mimic polytrauma inflammatory conditions by supplementing the culture medium with an inflammatory cocktail (polytrauma cocktail – PTC) or with or lipopolysaccharide (LPS) and nigericin, an antibiotic extracted from Streptomyces hygroscopicus and potent activator of the NLRP3 inflammasome. In cultured cardiomyocytes exposed to these inflammatory cocktails, some of them transfected with specific miRNA species identified in the clinical study, cell viability was tested with Alamar blue and caspase 1 (activator of pyroptosis) and caspase 3-7 (effectors of apoptosis) activity were tested via chemiluminescence assays. The tests were completed with a qRT-PCR assay for these miRNA species. The in vitro experiments confirmed involvement in myocardial pathophysiology of the miRNA markers identified in the clinical study. Specifically, miR-885-5p expression was increased by exposure to both inflammatory cocktails, while miR-122-5p was also increased but without reaching statistical significance. Other two miRNA species, miR-499a-5p and miR-194-5p, were found to play more important roles in apoptosis and inflammasome activation. Overall, the study was well conceived and executed, the analysis of experimental data was rigorous, leading to valuable conclusions that may be of interest to a wide readership and may result in development of new molecular markers potentially useful for clinical outcome prediction in thoracic trauma patients; therefore, I would recommend publication of the manuscript in the Cells journal.
Minor comment: it may be useful to specifiy that the cardiac-specific isoform of troponin T (cTnT) was measured in the clinical study, and not the skeletal-muscle-specific isoforms.
Comments on the Quality of English LanguageA revision of the manuscript by an experienced (preferably native) English speaker and minor corrections are necessary.
Round 2
Reviewer 1 Report
Comments and Suggestions for Authors
Revisions are appropriate and have made manuscript more relevant to discuss the roles of miRNAs as useful biomarker of myocardial injury after polytrauma.
Reviewer 2 Report
Comments and Suggestions for Authors
Authors have answered all comments point by point.